# 1,4-Aryl migration in ketene-derived enolates by a polar-radical-crossover cascade

Niklas Radhoff[1] & Armido Studer [1✉]

The arylation of carboxylic acid derivatives via Smiles rearrangement has gained great interest in recent years. Both radical and ionic approaches, as well as radical-polar crossover concepts, have been developed. In contrast, a reversed polar-radical crossover approach remains underexplored. Here we report a simple, efficient and scalable method for the preparation of sterically hindered and valuable α-quaternary amides via a polar-radical crossover-enolate oxidation-aryl migration pathway. A variety of easily accessible *N*-alkyl and *N*-arylsulfonamides are reacted with disubstituted ketenes to give the corresponding amide enolates, which undergo upon single electron transfer oxidation, a 1,4-aryl migration, desulfonylation, hydrogen atom transfer cascade to provide α-quaternary amides in good to excellent yields. Various mono- and di-substituted heteroatom-containing and polycyclic arenes engage in the aryl migration reaction. Functional group tolerance is excellent and substrates as well as reagents are readily available rendering the method broadly applicable.

[1] Organisch-Chemisches Institut, Westfälische Wilhelms-Universität, Corrensstrasse 40, 48149 Münster, Germany. ✉email: studer@uni-muenster.de

S ince their first description by Staudinger in 1905[1], ketenes have emerged as highly useful starting materials and inter-mediates in organic synthesis[2–9]. The high value of ketenes in synthesis lies in their ability to undergo clean and efficient [2 + 2] cycloadditions[10] with ketenes in dimerizations[11–15] and with ketones[16–21], imines[9,22,23] or alkenes[24,25] for the construction of four-membered and also larger ring structures[26–32]. These reactions often follow a stepwise mechanism with the addition of a nucleo-phile onto the ketene as the initial step. Equally important but underrepresented in synthesis is conversion of ketenes to the cor-responding enolates by nucleophilic attack of C[33,34]-, N[35,36]-, Si[37]- and O[38,39]-anions. This is particularly interesting for the generation of sterically demanding enolates with controlled *cis/trans*-stereo-chemistry. Notably, enolates of α,α-disubstituted amides are gen-erally not accessible via α-deprotonation. Nevertheless, the functionalization of enolates derived from α,α-disubstituted amides

is of high synthetic interest because the products, all-carbon α-quaternary amides, are important compounds in pharmaceutical chemistry as they often show biological activity for example as anti-nausea agents (e.g. Netupitant®) or spasmolytics[40]

In recent years, the concept of aryl migration[41–60] in general and especially for the α-arylation of both pre-functionalized and non-functionalized amides has gained great attention. Many years after the seminal works of Speckamp[61] and Motherwell[62] who pioneered the field of radical aryl migration from sulfur to car-bon, Nevado's group developed an approach to access α-quaternary amides comprising a conjugate radical addition to an acryl sulfonamide followed by radical 1,4-aryl migration onto an intermediately generated α-amide radical (Fig. 1a)[63–65]. Very recently, the same group published a stereoselective variant of this reaction in which enantiomerically pure acrylamides were used as starting materials that contain a chiral aryl sulfinyl group as a

a) **Radical** 1,4 aryl migration for the construction of a-quaternary amides
(*Nevado, Zhang/Wan, Studer*)

b) **Anionic** C-H-arylation by aryl migration (*Clayden*)

c) **Radical-polar crossover** followed by **anionic** 1,4-aryl migration (*Clayden*)

d) **Polar-radical crossover** followed by **radical** 1,4 aryl migration (*this work*)

**Fig. 1 Ionic and radical 1,4 aryl migration reactions—various strategies for C(sp²)-C(sp³) bond formation. a** Radical 1,4-aryl migration[63-68]. **b** Anionic migration[77]. **c** Radical-polar crossover migration[78]. **d** This work: Polar-radical crossover migration.

stereodirecting moiety which also serves as the aryl donor[66]. Along with radical addition to acrylates, such α-amide radical intermediates can also be generated via reductive C−X-bond cleavage of α-halo(sulfon)amides[67] or via intramolecular hydrogen atom transfer to aryl radicals by using *ortho*-iodoaryl sulfonamides as radical precursors[68]. Of note, the latter approach represents an α-C(sp³)−H arylation where the prefunctionalization of the α-position of the amide substrate is not required. However, the common limitation of all these methods lies in their restricted applicability towards formation of α,α-dialkyl-α-aryl amides, as they do not allow the efficient preparation of α,α-diaryl-α-alkyl amides.

Complementary to radical aryl migrations, anionic 1,4-aryl migrations can also be applied for C(sp²)−C(sp³) bond formation. Dohmori and coworkers observed anionic aryl migration in sulfonamide enolates as early as the 1950s[69–76]. More recently, Clayden's group developed a valuable method for the stereoselective arylation of the α-C(sp³)−H bond in imidazolidinones (Fig. 1b)[77]. These reactions proceed via deprotonation (enolate formation) and subsequent anionic 1,4-aryl migration from nitrogen to carbon. The same group recently introduced an elegant method for the preparation of α-arylated amines via radical-polar crossover (Fig. 1c)[78]. The cascade is initiated by a radical addition to an eneurea with subsequent reduction of the adduct radical to the corresponding anion followed by ionic 1,4-aryl migration. The switching from the radical to the anionic mode in such cross-over processes may offer advantages and will also open new opportunities in reaction design.

Keeping the crossover benefits in mind, we herein present a reversed polar-radical crossover strategy that uses an anionic addition as the initial step of the cascade with subsequent SET-oxidation and radical translocation, harvesting the great potential of the radical aryl migration reaction. As substrates, we are using ketenes that can readily be prepared by established methodology in a one-pot sequence via acid chlorides easily obtained from commercially available carboxylic acids. As discussed above, ketenes serve as efficient enolate precursors. Hence, the addition of a lithiated arylsulfonamide to a ketene leads to the corresponding sterically demanding α-alkyl-α-aryl or α,α-dialkyl amide enolate (Fig. 1d). The polar-radical crossover is achieved by enolate SET-oxidation which is followed by a radical 1,4-aryl migration. SO₂-extrusion and hydrogen atom abstraction will complete the sequence. Of note, the sulfonamides are prepared in high yield in an easy and scalable reaction from inexpensive amines and aryl sulfonyl chlorides, which are commercially available in great variety. Therefore, this process allows to prepare a large number of all carbon α-quaternary amides in a simple one-pot process starting from easily accessible precursors.

## Results and discussion

**Optimization study.** We chose N-isopropylbenzenesulfonamide **1a** and ethyl phenyl ketene **2a** as model substrates for reaction optimization. The sulfonamide **1a** was first deprotonated with *n*-BuLi (1.1 eq.) in THF and then mixed with the preformed ketene **2a**. SET oxidation of the enolate was initially attempted by irradiation (456 and 467 nm) in the presence of a photoredox catalyst, [Ir(dF(CF₃)ppy)₂(dtbpy)]PF₆ (5 mol%) or [Ru(bpy)₃](PF₆)₂ (5 mol%) (Table 1, entry 1 and 2). Disappointingly, the desired α-arylated amide **3aa** was not formed in both cases. We next tested

---

**Table 1 Reaction optimization[a,b].**

| Entry | Oxidant (equiv.) | Base (equiv.) | Solvent (M) | Yield |
|---|---|---|---|---|
| 1 | [Ir(dF-CF₃)ppy)](dtbpy)]PF₆ (0.05) | *n*-BuLi (1.1) | THF (0.1) | n.d. |
| 2 | [Ru(bpy)₃](PF₆)₂ (0.05) | *n*-BuLi (1.1) | THF (0.1) | n.d. |
| 3 | CuCl₂ (1.2) | *n*-BuLi (1.1) | THF (0.1) | 22%[c] |
| 4 | FcBF₄ (1.2) | *n*-BuLi (1.1) | THF (0.1) | 20%[c] |
| 5 | CuCl₂ (1.2) | *n*-BuLi (1.1) | THF (0.01) | 38% |
| 6 | FcBF₄ (1.2) | *n*-BuLi (1.1) | THF (0.01) | 60% |
| 7[d] | FcBF₄ (1.0) | *n*-BuLi (1.1) | THF (0.01) | 61% |
| 8[d] | FcBF₄ (0.3) | *n*-BuLi (1.1) | THF (0.01) | Traces |
| 9[d] | FcBF₄ (0.1) | *n*-BuLi (1.1) | THF (0.01) | n.d. |
| 10[d] | FcBF₄ (1.0) | *n*-BuLi (1.1) | PhMe (0.01) | n.d. |
| 11[d] | FcBF₄ (1.0) | *n*-BuLi (1.1) | PhH (0.01) | 32% |
| **12[d]** | **FcBF₄ (1.0)** | **_n_-BuLi (1.1)** | **Et₂O (0.01)** | **>99% (94%[c])** |
| 13[d] | None | *n*-BuLi (1.1) | Et₂O (0.01) | n.d. |
| 14[d] | FcBF₄ (1.0) | None | Et₂O (0.01) | n.d. |
| 15[d] | FcBF₄ (1.0) | NaH (1.1) | Et₂O (0.01) | n.d. |
| 16[d] | FcBF₄ (1.0) | DBU (1.1) | Et₂O (0.01) | n.d. |
| 17[d] | FcBF₄ (1.0) | LiHMDS (1.1) | Et₂O (0.01) | 81% |
| 18[d] | *fac*-[Ir(ppy)₃] (0.05) | *n*-BuLi (1.1) | Et₂O (0.01) | n.d. |
| 19[d] | Mes-Acr⁺ClO₄⁻ (0.05) | *n*-BuLi (1.1) | Et₂O (0.01) | n.d. |
| 20[d] | [Ru(bpz)₃](PF₆)₂ (0.05) | *n*-BuLi (1.1) | Et₂O (0.01) | n.d. |
| 21[d] | [Ir(ppy)₂(dtbpy)]PF₆ (0.05) | *n*-BuLi (1.1) | Et₂O (0.01) | n.d. |
| 22[d] | 4CzIPN (0.05) | *n*-BuLi (1.1) | Et₂O (0.01) | n.d. |

[a]Reactions (0.200 mmol) were performed under an atmosphere of argon. n.d. = not detected (GC-MS). FcBF₄ = Ferrocenium tetrafluoroborate. DBU = 1,8-Diazabicyclo[5.4.0]undec-7-en. LiHMDS = Lithium bis(trimethylsilyl)amide. Mes-Acr⁺ClO₄⁻ = 9-Mesityl-10-methylacridinium perchlorate. 4CzIPN = 2,4,5,6-tetra(carbazole-9-yl)isophtalonitrile. The optimized reaction conditions are marked in bold.
[b]Yields determined by GC-FID using *n*-dodecane as internal standard.
[c]Isolated yield.
[d]Ketene **2a** in Et₂O (0.06 M) added by syringe pump over 30 min.

**Table 2 Reaction optimization using catalytic amounts of ferrocenium tetrafluoroborate[a,b].**

| Entry | Co-oxidant (equiv.) | Additive (equiv.) | c (M) | Yield |
|---|---|---|---|---|
| 1[c] | fac-[Ir(ppy)₃] (0.05)[j] | None | 0.01 | <5% |
| 2[c] | Mes-Acr⁺ClO₄⁻ (0.05)[j] | None | 0.01 | Traces |
| 3[c] | [Ru(bpz)₃](PF₆)₂ (0.05)[j] | None | 0.01 | Traces |
| 4[c] | [Ru(bpy)₃](PF₆)₂ (0.05)[j] | None | 0.01 | 5% |
| 5[c] | [Ir(ppy)₂(dtbpy)]PF₆ (0.05)[j] | None | 0.01 | 8% |
| 6[c] | 4CzIPN (0.05)[j] | None | 0.01 | Traces |
| 7[c] | [Ir(dF-CF₃)ppy)₂(dtbpy)]PF₆ (0.05)[j] | None | 0.01 | 39% |
| 8[d] | [Ir(dF-CF₃)ppy)₂(dtbpy)]PF₆ (0.05)[j] | None | 0.01 | 48% |
| 9[d,e] | [Ir(dF-CF₃)ppy)₂(dtbpy)]PF₆ (0.01)[j] | None | 0.01 | 25% |
| 10[d,f] | [Ir(dF-CF₃)ppy)₂(dtbpy)]PF₆ (0.05)[j] | None | 0.01 | 45% (40%[g]) |
| 11[c] | [Ir(dF-CF₃)ppy)₂(dtbpy)]PF₆ (0.05)[j] | BrCCl₃ (3.0) | 0.01 | 13% |
| 12[c] | [Ir(dF-CF₃)ppy)₂(dtbpy)]PF₆ (0.05)[j] | K₂S₂O₈ (3.0) | 0.01 | 23% |
| 13[c] | [Ir(dF-CF₃)ppy)₂(dtbpy)]PF₆ (0.05)[j] | TBHP (3.0) | 0.01 | 23% |
| 14[c] | [Ir(dF-CF₃)ppy)₂(dtbpy)]PF₆ (0.05)[j] | Bu₃SnSnBu₃ (1.2) | 0.01 | <5% |
| 15[c] | [Ir(dF-CF₃)ppy)₂(dtbpy)]PF₆ (0.05)[j] | TMS₃SiH (1.2) | 0.01 | Traces |
| 16[c] | 1,4-benzoquinone (1.0) | None | 0.01 | Traces |
| 17[c] | p-chloranil (1.0) | None | 0.01 | 5% |
| 18[c] | DDQ (1.0) | None | 0.01 | 48% |
| 19[c] | Bobbitt's salt (1.0) | None | 0.01 | 29% |
| 20[c] | TEMPO-BF₄ (1.0) | None | 0.01 | 61% |
| 21 | TEMPO-BF₄ (1.0) | None | 0.04 | 64% |
| 22[e] | TEMPO-BF₄ (1.0) | None | 0.04 | 11% |
| 23[h] | TEMPO-BF₄ (1.0) | None | 0.04 | 56% |
| 24 | TEMPO-BF₄ (2.0) | None | 0.04 | 54% |
| 25[i] | TEMPO-BF₄ (1.0) | None | 0.04 | n.d. |

[a]Reactions (0.200 mmol) were performed under an atmosphere of argon. n.d. = not detected (GC-MS). FcBF₄ = Ferrocenium tetrafluoroborate. Mes-Acr⁺ClO₄⁻ = 9-Mesityl-10-methylacridinium perchlorate. 4CzIPN = 2,4,5,6-tetra(carbazole-9-yl)isophthalonitrile. TBHP = tert-Butyl hydroperoxide. Bobbitt's salt = 4-(Acetylamino)-2,2,6,6-tetramethyl-1-oxo-piperidinium tetrafluoroborate. DDQ = 4,5-Dichloro-3,6-dioxocyclohexa-1,4-diene-1,2-dicarbonitrile. TEMPO-BF₄ = 2,2,6,6-Tetramethyl-1-oxo-piperidinium tetrafluoroborate.
[b]Yields determined by GC-FID using n-dodecane as internal standard.
[c]Ketene 2a in Et₂O (0.06 M) added by syringe pump over 30 min.
[d]Ketene 2a in Et₂O (0.06 M) added by syringe pump over 120 min.
[e]10 mol% FcBF₄.
[f]30 mol% FcBF₄.
[g]Isolated yield.
[h]40 mol% FcBF₄.
[i]Without FcBF₄.
[j]Irradiation with blue LED (467 nm).

the SET-oxidation of the intermediate enolate with stoichiometric CuCl₂ (1.2 eq.) and ferrocenium tetrafluoroborate (FcBF₄, 1.2 eq.). Pleasingly, the targeted amide **3aa** was obtained in encouraging 22% and 20% yield, respectively (Table 1, entry 3 and 4). As a side product, the ketene dimer was observed in these transformations. To suppress dimer formation, the concentration of the reaction mixture was decreased to 0.01 M and yield improved to 38% with CuCl₂ (1.2 eq.) and further to 60% with FcBF₄ (1.2 eq.) (Table 1, entry 5 and 6). When the ketene is slowly added as a solution in THF (0.06 M) over half an hour by syringe pump, the yield minimally increased to 61% with 1.0 eq. of FcBF₄ (Table 1, entry 7). By using a sub-stoichiometric amount FcBF₄, only traces of the product **3aa** were obtained (Table 1, entry 8 and 9). Next, the influence of the solvent was investigated and the desired **3aa** was not formed in toluene, whereas in benzene the yield decreased to 32% (Table 1, entry 10 and 11). To our delight, in diethyl ether under otherwise identical conditions, a quantitative conversion was achieved, as analyzed by GC-FID and the amide **3aa** was isolated in an excellent 94% yield (Table 1, entry 12). The reduced oxidant, ferrocene, which can readily be re-oxidized to the ferrocenium ion, was recovered in 96% yield. In a control experiment in the absence of oxidant, product **3aa** was not identified, which indicates the radical nature of the aryl migration excluding an anionic arylation pathway (Table 1, entry 13). A further control experiment in absence of base also resulted

in no product formation (Table 1, entry 14). Substitution of n-BuLi with milder and non-nucleophilic bases such as NaH and DBU also shuts down the reaction (Table 1, entry 15 and 16). Only with LiHMDS was product formation observed, but with a reduced yield of 81% (Table 1, entry 17), suggesting that the Li-enolate is a crucial intermediate in the cascade described herein.

Even though the inexpensive and commercially available stoichiometric FcBF₄ can be near quantitatively recovered and subsequently recycled, further attempts were made to develop a variant of the reaction using catalytic amounts of an oxidant. Initial screenings using photoredox catalysis in the absence of any co-oxidants (Table 1, entry 1 and 2) failed. Further addressing such a process, we tested other photocatalysts with both stronger reducing and oxidizing power, but none of them led to the formation of the desired product (Table 1, entries 18–22). We then envisioned to establish a dual catalysis cycle using photoredox catalysis to in situ recycle the ferrocenium oxidant. The intermediate amidyl radical formed after the aryl migration sequence might regenerate the photoredox catalyst under formation of the corresponding N-anion. Various photocatalysts in combination with 20 mol% of the ferrocenium oxidant were screened and a maximum yield of 8% was achieved (Table 2, entries 1–5). However, with [Ir(dF(CF₃) ppy)₂(dtbpy)]PF₆ (5 mol%), FcBF₄ (20 mol%) under irradiation with a blue LED (467 nm) the desired amide **3aa** was formed in encouraging 39% yield (Table 2, entry 7). When ketene **2a** was

**Fig. 2 Variation of the *N*-substituent in the arylsulfonamide.** Reaction scale: 0.200 mmol. n.d. = not detected. d.r. = diasteromeric ratio. FcBF$_4$ = Ferrocenium tetrafluoroborate. Ketene **2a** in Et$_2$O (0.06 M) over 30 min by syringe pump. [a]LiCl (7.0 eq.) was added.

added over a period of two hours by syringe pump the yield was further improved to 48% (Table 2, entry 8). Lowering the catalyst loading of the FcBF$_4$ led to a reduced yield (Table 2, entry 9), and increasing the amount of FcBF$_4$ to 30 mol% also did not affect the reaction outcome (Table 2, entry 10). We next tried stochiometric co-oxidants as additives (BrCCl$_3$, *tert*-butyl hydroperoxide (TBHP) and K$_2$S$_2$O$_8$) but were not able to further increase the yield (Table 2, entries 11–13). The addition of (TMS)$_3$SiH or Bu$_3$SnSnBu$_3$ as H-atom donors or trapping reagents for the amidyl radical to form silyl or stannyl radicals, respectively, which can subsequently be reduced by the photoredox catalyst to close the catalysis cycle, did also not increase the yield of **3aa** (Table 2, entries 14 and 15).

As an alternative approach, we replaced the photoredox catalyst with cheap stochiometric co-oxidants. Thus, with *p*-chloranil (1.0 equiv.) and 1,4-benzoquinone (1.0 equiv.) in combination with FcBF$_4$ (20 mol%), the desired amide **3aa** was detected in trace amounts only (GC-MS, Table 2, entries 16 and 17). Stronger oxidants such as DDQ (48%), Bobbitt's salt (21%) and TEMPO-BF$_4$ (61%) gave significantly better yields (Table 2, entries 18–20). The best result was obtained, when TEMPO-BF$_4$ (1.0 equiv.) and FcBF$_4$ (20 mol%) were reacted in Et$_2$O (0.04 M) with ketene **2a** being added in one portion. In this case, amide **3aa** was formed in 64% yield (Table 2, entry 21). Lowering the loading of FcBF$_4$ to 10 mol% led to a reduced yield (11%), whereas increasing the iron(III) loading to 30 mol% or the amount of co-oxidant to 2.0 equiv. also gave slightly reduced yields of 54% and 56%, respectively (Table 2, entries 22–24). A control experiment without FcBF$_4$ showed no product formation, indicating that TEMPO-BF$_4$ is not able to directly oxidize the

enolate (Table 2, entry 25). Since the catalytic variants developed provided lower yields of the targeted aryl migration product **3aa**, we decided to run the scope study applying the procedure that uses the cheap stoichiometric recyclable FcBF$_4$ as the oxidant.

**Substrate scope.** We first investigated the scalability of the method. For this purpose, the model substrate **1a** and ethyl phenyl ketene **2a** were converted to the amide **3aa** under the optimized conditions on a 5.0 mmol scale. Compared to the small-scale experiment (0.2 mmol), a comparatively excellent yield of 91% (1.28 g) was obtained (Fig. 2). Next, we investigated the substrate scope keeping **2a** as the ketene component. First, different substituents on the N-atom of the benzenesulfonamide were investigated. In this series, *N*-alkylbenzene-sulfonamides **1a-f** were found to be the most efficient substrates and bulkier alkyl substituents on the N-atom lead to higher yields most likely due to conformational effects. Thus, for the sterically least demanding *N*-methyl derivative **3ab** a yield of 57% was noted, whereas the *N*-benzyl and *N*-trifluoroethyl derivatives **3ac** and **3ad** gave a 78% yield each. In case of the *N*-methyl derivative **3ab** protonated enolate was observed as the side product. The best result (98% yield) was obtained for the *N*-cyclohexyl derivative **3ae** and the *N*-dioxanyl derivative **3af** could be isolated in 54% yield. The substrate bearing the sterically demanding *N*-tert-butyl group provided the aryl migration product **3ag** in only 33% yield and unreacted sulfonamide **1g** was observed. Thus, the high steric demand of the *tert*-butyl substituent negatively influences initial enolate formation. *N*-Aryl amides also engage in this cascade,

**Fig. 3 Variation of migrating aryl group.** Reaction scale: 0.200 mmol. Ketene **2a** in Et$_2$O (0.06 M) over 30 min by syringe pump. FcBF$_4$ = Ferrocenium tetrafluoroborate.

albeit with lower efficiency, as documented by the preparation of **3ah** (26%).

Next, it was investigated whether a diastereoselective aryl migration is feasible by using enolates generated with chiral lithium amides. To this end, *p*-tolylsulfonamides derived from (*S*)-phenethylamine and valine methyl ester were reacted under the optimized conditions. The corresponding amides **3ai** and **3aj** were isolated in high yields (78–80%) as a 1:1 mixture of the two diastereoisomers. Guided by the Myers alkylation[79–81], *p*-tolylsulfonamides **1k-n** derived from *pseudo*-norephenamine were prepared and subjected to the aryl migration sequence in combination with ketene **2a**. However, reaction did not work for the O-unprotected compound (**3ak**) and for the O-protected congeners, diastereoselectivity could not be controlled (**3al-an**).

Studies were continued by investigating the substrate scope with respect to the migrating aryl group with **2a** as the ketene component. *p*-Methyl, *p*-fluoro and *p*-chloro substituents are tolerated on the migrating aryl moiety and the corresponding amides were isolated in 70% (**3ao**), 63% (**3ap**) and 84% (**3aq**) yield (Fig. 3). The method tolerates both electron-deficient and electron-rich arenes as migrating units and the *p*-CF$_3$- and *p*-methoxy-phenyl substituted amides **3ar** and **3as** were isolated in 87% and 69% yield. Lower yields were noted for the *p*-nitro, *p*-cyano, and *p*-amido congeners (see **3at, 3au,** and **3av**). Substituents in *ortho* and *meta* position on the migrating

aryl moiety are also tolerated (**3aw**, 67%; **3ax**, 76%; **3ay**, 49%). Reaction with the sterically hindered 1-naphthylsulfonamide was less efficient and the product **3az** was isolated in 19% yield, but the 2-naphthyl congener **3aaa** was obtained with 73% yield. The migration of heteroarenes was found to be lower yielding. Thus, 2-thienyl (**3aab**), 2-benzothienyl (**3aac**), and 2-benzofuryl (**3aad**) amides were obtained in 27–43% yield. The pyridyl and benzothiazolyl migration products **3aae** and **3aaf** were formed in trace amounts only, as detected by ESI-MS.

Finally, different ketenes were tested and **1a** was selected as the sulfonamide component in this series (Fig. 4). Starting with alkyl phenyl ketenes **2b-d**, the methyl-, isopropyl- and cyclopentyl-phenyl amides **3ba-da** were obtained in 29–63% yields. A tetrahydronaphthyl derivative could also be accessed (see **3ea**, 42%) and the *p*-iodo- **3fa** as well as the *p*-bromophenylamide **3ga** were successfully prepared by using ethyl *p*-iodophenylketene **2f** and ethyl *p*-bromophenylketene **2g** as precursors. Moreover, the ibuprofen® derivative **3ha** was isolated in good yield (75%). However, diphenyl ketene **2i** is not a suitable substrate and the corresponding amide **3ia** could be detected in traces only by ESI-MS. The higher stability and the larger steric demand of the α,α-diphenylated α-amide radical likely prevents the aryl migration. While most alkyl aryl ketenes are rather stable compounds that

**Fig. 4 Variation of the ketene.** Reaction scale: 0.200 mmol. Ketene **2** in Et$_2$O (0.06 M) over 30 min by syringe pump. FcBF$_4$ = Ferrocenium tetrafluoroborate.

can be stored in the freezer under argon atmosphere for some time, dialkyl ketenes are reactive intermediates that have to be used in situ directly after their generation. We were pleased to find that dialkyl ketenes could also be successfully applied to the enolate formation aryl migration cascade. For example, the methyl benzyl derivative **3ja** was obtained in 49% yield. Better yields were noted for the methyl cyclohexylamide **3ka** and the cycloheptane derivative **3la** (74–81%).

We made the experience that amide hydrolysis in our case was very challenging. Addressing that issue, amide **3af** with the assistance of the hydroxyl groups liberated after acetal hydrolysis was readily hydrolyzed under acidic conditions to the corresponding α-quaternary carboxylic acid **4** in 71% yield (Fig. 5). Thus, with this strategy a variety of α-quaternary carboxylic acids can be prepared, further improving the applicability of the introduced method.

**Reaction mechanism**. Our suggested mechanism for the polar-radical crossover cascade is presented in Fig. 6. First, the sulfonamide **1a** is deprotonated with *n*-BuLi and the resulting Li-amide **1a**-Li then adds to the ketene to generate the enolate **A**. The enolate **A** is oxidized by the ferrocenium ion (Fe$^{III}$, E$_{1/2}$ (Fc/Fc$^+$) = 0.32 V vs. SCE[82]) to generate the α-amide radical **B**[83–90], which attacks the arene at the *ipso*-position to give the spirocyclic intermediate **C**. Homolytic cleavage of the C−S bond and extrusion of SO$_2$ lead to the amidyl radical **D**[64,67,68]. Since ferrocene (Fe$^{II}$) is obtained as a by-product in stoichiometric amounts, the amidyl radical **D** is not efficiently reduced by ferrocene preventing the Fe-salt to act as a redox catalyst. Therefore, we assume that the amidyl radical **D** further reacts via hydrogen abstraction[67] from Et$_2$O to finally afford product amide **3**. This may also explain why ethereal solvents provide the highest yields.

**Fig. 5 Follow-up chemistry.** Acidic hydrolysis of amide **3af**.

In summary, we presented a simple, efficient and scalable method for the preparation of sterically hindered and synthetically valuable α-quaternary amides via a polar-radical crossover enolate oxidation-aryl migration sequence. Ketenes were introduced as highly valuable precursors for the generation of α-amide radicals. The starting materials, both the ketenes and the arylsulfonamides, are easily prepared from a wide variety of commercially available compounds. More than thirty N-alkyl as well as N-arylamides bearing an all carbon α-quaternary center were readily prepared via this cascade, convincingly documenting the broad applicability of the method and also showing the excellent functional group tolerance. The method convincingly shows the potential of polar-radical crossover processes in organic synthesis by merging valuable anionic with equally important radical steps.

## Methods

**Representative procedure for the arylation in ketene-derived enolates**. To a Schlenk tube were added sulfonamide **1a** (39.9 mg, 0.200 mmol, 1.0 eq.) and anhydrous Et$_2$O (15 mL). *n*-BuLi (1.6 M in hexanes, 138 µL, 0.220 mmol, 1.1 eq.)) was added and the mixture was stirred for 30 min at room temperature. Ferrocenium tetrafluoroborate (54.7 mg, 0.200 mmol, 1.0 eq.) was added, and stirring

**Fig. 6 Plausible mechanism of the polar-radical crossover cascade.** The reaction proceeds through anionic (**1a-Li** and **A**) and radical intermediates (**B**, **C** and **D**).

continued for further 15 min. A solution of ketene **2a** (39 μL, 0.30 mmol, 1.5 eq.) in anhydrous Et$_2$O (5 mL) was added over 30 min by syringe pump. After stirring at room temperature overnight, the reaction mixture was concentrated and subjected to flash column chromatography (pentane) to recover ferrocene (35.7 mg, 0.192 mmol, 96%). Flushing the column with EtOAc and subsequent purification by RP-MPLC (MeOH/H$_2$O, gradient from 20% to 90%) led to isolation of α-quaternary amide **3aa** as a colorless solid (52.9 mg, 0.188 mmol, 94%). For details on ferrocenium tetrafluoroborate recycling, see the Supplementary Information.

## Data availability

Supplementary information and chemical compound information accompany this paper at www.nature.com/ncomms. The data supporting the results of this work are included in this paper or in the Supplementary Information and are also available upon request from the corresponding author.

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

## Acknowledgements
We thank the WWU Münster and the European Research Council ERC (advanced grant agreement No. 692640, awarded to A.S.) for supporting this work.

## Author contributions
N.R. and A.S. conceived and designed the experiments. N.R. performed the experiments and analyzed the data. N.R. and A.S. wrote the manuscript.

## Funding

## Competing interests
The authors declare no competing interests.
