## [Peer Review File · Nature Communications]

REVIEWER COMMENTS

Reviewer #1 (Remarks to the Author):

In this manuscript, Studer and Radhoff have described a polar-radical crossover reaction for the synthesis of useful α -quaternary amides. The reaction of sulfonamide with ketene gives rise to amide enolates, which proceeds via single electron oxidation by ferrocenium salt, 1,4-aryl migration, desulfonylation, hydrogen atom transfer cascade to give α -quaternary amides in moderate to good yields. The substrate scope is reasonably broad, and the manuscript is well written. One weak point of the reaction is the use of stoichiometric amounts of ferrocenium salt. A redox neutral process could further increase the synthetic value of the method. Complementary to radical migration, polar-radical/radical-polar crossover processes open new opportunities in reaction design. Though radical Smiles rearrangement is well-established, the overall transformation herein, in particular the oxidation of enolate to initiate radical, is interesting. Upon consideration of the novelty and the potential utility of the method, this reviewer supports its acceptance in NatComm after revisions.

1. Ketenes are prepared from acyl chlorides, and are usually sensitive to moisture and nucleophiles. If avoiding the generation of ketenes and instead making acyl sulfonamides from the same acyl chlorides, and then subjecting the acyl sulfonamides to the same procedures (reacting with BuLi to generate enolate followed by ferrocenium oxidation to generate radical), would the reaction result in similar results?
2. Scheme 3, the migration of thienyl and pyridyl group is not good. According to reference (Org. Lett. 2020, 22, 5947), the migration aptitude of these two groups among heteroaryls are relatively lower. Have the authors tried the migration of benzofuryl, benzothiazolyl, or benzothienyl? They should give better yields.
3. It is surprising that nitro and cyano groups are tolerated in presence of BuLi but deliver low yields. What are the byproducts according to mass balance?
4. Some closely related papers on the aryl migration from S to C are missed, e.g. Chem. Soc. Rev. 2021, 50, 11577 (review); ACIE 018, 57, 17156; ACIE 2020, 59, 8195; ACIE 2021, 60, 3714, etc.

Reviewer #2 (Remarks to the Author):

The paper describes the migration of aromatic rings to generate quaternary centres initiated by oxidation of an enolate to a radical. Similar aryl transfers have been reported by other groups (Nevado, Clayden), but the unique aspect here is the fact that the reaction proceeds by a polar step followed by a radical step as opposed to the reverse (or entirely polar/entirely radical - these possibilities are outlined in scheme 1). The method is sound, though somewhat limited by the stability of the ketene starting materials, and in many cases the yields are moderate. The biggest hurdle to applicability is the use of stoichiometric ferrocenium oxidant – photoredox catalysts failed. The products are racemic. Overall this is an interesting reaction, but with several drawbacks with regards to utility, and the case is not made for publication in Nature Comms.

Referee report

The authors reported a simple, efficient and scalable method for the preparation α -quaternary amides via a polar-radical crossover-enolate oxidation-aryl migration pathway. The reaction of lithium salt *N*-arylsulfonamides with ketenes to give amide enolates, and the single electron transfer oxidation of the enolates with FcBF_4 are known methods. The radical 1,4-migration to construct quaternary amides were reported in recent years. The referee suggest the paper published as a full paper rather than a communication.

Other comments:

In Table 1, the authors reported two cases of photoinduced oxidation with negative results. Other photocatalysts with higher oxidation ability may abstract an electron to generate a radical.

In Table 1, when stoichiometric amount of oxidant was applied, reasonable yield was obtained, while with substoichiometric amounts of oxidant, the yield is trace amount (entry 6 vs entry 8).

Why?

As is claimed by the authors, a bulky substituent on N atom favors the reaction, why not try *t*-Butyl sulfonamide?

Reply to the evaluation by the Reviewer 1

- 1.) One weak point of the reaction is the use of stoichiometric amounts of ferrocenium salt. A redox neutral process could further increase the synthetic value of the method.

Answer: We agree with Reviewer 1 that a redox neutral process requiring only substoichiometric amounts of iron(III) would further improve the value of this work. If the amidyl radical can be reduced to the corresponding N-anion, a redox neutral process would theoretically be realized. However, it is well known that this particular reduction is difficult to achieve and often competes with hydrogen abstraction from the solvent, terminating the catalytic cycle. Moreover, it is not clear whether SO₂ fragmentation (depends on the N-substituent) is fast enough in all cases and thus reduction of the aminosulfonyl radical must be considered as well. We tried intensively to establish a catalytic redox neutral process, as detailed below (see new entries in Table 1 and new Table 2):

First, we investigated various photoredox catalysts (5 mol%) that have a higher or similar oxidation potential compared to ferrocenium tetrafluoroborate. In addition, we also investigated photoredox catalysts known to be highly reducing, such as *fac*-Ir(ppy)₃. Unfortunately, none of these catalysts provided any of the targeted aryl migration product. As unwanted side reaction, we found that the enolate decomposes rapidly in an intramolecular proton transfer reaction. The proton is abstracted by the enolate intramolecularly from the ortho-position of the aryl group. The aryl anion thus generated then cyclizes onto the amide carbonyl group eventually resulting in an intramolecular acyl migration. This deprotonation/acyl migration sequence competes with the enolate oxidation and thus a high concentration of oxidant is required, rendering a catalytic process highly challenging.

Since ferrocenium tetrafluoroborate efficiently oxidizes the enolate, as shown in the stoichiometric experiments, we also considered re-oxidation of ferrocene in situ by a photoredox catalyst. The reduced state of the photoredox catalyst could then potentially reduce the intermediate amidyl radical (or aminosulfonyl radical), thus closing the photoredox catalysis cycle. Various redox catalysts were tested along those lines and by using 20 mol% iron(III) and 5 mol% of Ir(((dF-CF₃)ppy)₂dtbpy]PF₆, the desired alpha-arylated amide was obtained in reasonable 48% yield (see Table 2, entry 8). Other photoredox catalysts tested did not work. Unfortunately, the yield could not be further increased by varying the amount of iron(III) and photocatalyst. Since efficient regeneration of the photocatalyst is critical for this reaction to work, we considered adding an external stoichiometric co-oxidant such as K₂S₂O₈, TBHP, or BrCCl₃. None of these additives increased the yield of amide **3aa**. We also tried TTMSS and hexabutylidistannane as additives that can react with the amidyl radical to form a silyl or stannyl radical that should then be reduced to regenerate the initial oxidation state of the photocatalyst, but again without success in terms of the yield of **3aa**.

Since the photoredox catalysis route developed could not compete with the stoichiometric variant, we next considered using catalytic amounts of ferrocenium salt in combination with a stoichiometric transition metal-free co-oxidant. *p*-Chloranil and 1,4-benzoquinone were found to be not suitable as co-oxidants, but with *Bobbitt's* salt, DDQ, and TEMPO-BF₄ in combination with FcBF₄ (20 mol%) in Et₂O (0.04 M), the desired amide **3aa** was indeed observed in yields up to 64% (Table 2, entry 21). However, the yield could not be further improved by systematically varying the reaction conditions. Neither a higher loading of the iron catalyst nor of the stoichiometric co-oxidant led to higher yields. A lower catalyst loading led to a drastic decrease in yield.

Since a catalytic variant of the reaction described here in various approaches does not give results nearly as good as the process that uses a stoichiometric amount of ferrocenium tetrafluoroborate,

we remain convinced that the latter is an extremely efficient and synthetically highly useful process. Especially considering that the stoichiometric oxidant is cheap, commercially available and very easy to recycle after use, as shown in the paper. The “catalytic” process delivers lower yields and are overall also more expensive, in particular considering the use of the expensive Ir-based photoredox catalyst.

2.) Ketenes are prepared from acyl chlorides, and are usually sensitive to moisture and nucleophiles. If avoiding the generation of ketenes and instead making acyl sulfonamides from the same acyl chlorides, and then subjecting the acyl sulfonamides to the same procedures (reacting with BuLi to generate enolate followed by ferrocenium oxidation to generate radical), would the reaction result in similar results?

Answer: We agree that this would be a very good alternative to access the same enolates. However, apart from the fact that the synthesis of sterically demanding *N*-acylated sulfonamides is synthetically sometimes challenging, in general *N,N*-disubstituted alpha-branched amides are usually not convertible to their enolates *via* alpha-deprotonation. Due to the allylic A(1,3)-strain, the alpha-proton is kinetically not acidic (conformationally fixed). Nevertheless, we synthesized the *N*-acylated sulfonamide **5** and attempted to convert it to the enolate using LDA as base, followed by oxidation with ferrocenium tetrafluoroborate. As expected, the desired arylation product **3aa** was not observed. We have mentioned in the main text that deprotonation to generate such enolates is difficult.

3.) Scheme 3, the migration of thienyl and pyridyl group is not good. According to reference (Org. Lett. 2020, 22, 5947), the migration aptitude of these two groups among heteroaryls are relatively lower. Have the authors tried the migration of benzofuryl, benzothiazolyl, or benzothieryl? They should give better yields.

Answer: We thank the referee for this useful hint. Indeed, the migratory aptitude of those fused heterocycles should be higher than for the corresponding monocycles. We synthesized the suggested benzothieryl, benzofuryl and benzothiazolyl sulfonamides and applied them as substrates in our reaction. In contrast to many other radical aryl migration reactions, lower yields were achieved for these heteroaryl systems. The benzothieryl and benzofuryl congeners **3aac** and **3aad** were isolated with 43% and 20%, respectively. Unfortunately, benzothiazolyl derived amide **3aaf** was only formed in trace amounts (ESI-MS). These examples were added to Scheme 3.

4.) It is surprising that nitro and cyano groups are tolerated in presence of BuLi but deliver low yields. What are the byproducts according to mass balance?

Answer: As Reviewer 1, we were also surprised that these functional groups are tolerated by our method. We believe that the deprotonation is far faster than reaction with the nitrile or nitro group and only a small excess of BuLi is used in these transformations. In these reactions, beside the desired arylation product, no distinct by-products were isolated in neither of those two reactions. However, trace amounts of side-products that derive from nucleophilic addition of BuLi to the nitrile or the nitro group were identified by ESI-MS.

5.) Some closely related papers on the aryl migration from S to C are missed, e.g. Chem. Soc. Rev. 2021, 50, 11577 (review); ACIE 018, 57, 17156; ACIE 2020, 59, 8195; ACIE 2021, 60, 3714, etc.

Answer: We thank the reviewer for pointing this out. The missing literature is now cited in the introduction.

Reply to the evaluation by the Reviewer 2

- 1.) The paper describes the migration of aromatic rings to generate quaternary centres initiated by oxidation of an enolate to a radical. Similar aryl transfers have been reported by other groups (Nevado, Clayden), but the unique aspect here is the fact that the reaction proceeds by a polar step followed by a radical step as opposed to the reverse (or entirely polar/entirely radical - these possibilities are outlined in scheme 1). The method is sound, though somewhat limited by the stability of the ketene starting materials, and in many cases the yields are moderate. The biggest hurdle to applicability is the use of stoichiometric ferrocenium oxidant – photoredox catalysts failed. The products are racemic. Overall this is an interesting reaction, but with several drawbacks with regards to utility, and the case is not made for publication in Nature Comms.

Answer: We thank Reviewer 2 for his comments. We have intensively tried to address the need of stoichiometric ferrocenium oxidant but failed using photoredox catalysis (Table 2). We refer to the comments addressed to referee 1 regarding that issue.

Reply to the evaluation by the Reviewer 3

- 1.) In Table 1, the authors reported two cases of photoinduced oxidation with negative results. Other photocatalysts with higher oxidation ability may abstract an electron to generate a radical.

Answer: We thank the reviewer for this suggestion. We have screened a small library of other photoredox catalysts during the course of this revision that have a higher or similar oxidation potential than ferrocenium tetrafluoroborate. Furthermore, with *fac*-Ir(ppy)₃, among others, also catalysts which are strongly reducing were tested. None of these catalysts led to product formation. In our opinion, this is due to the fact that the enolate decomposes as a competitive reaction to the desired SET oxidation. Thus, SET oxidation must occur fast (prior to enolate decomposition) and therefore a high concentration of oxidant is necessary to realize these cascades (difficult with a redox system present in catalytic amounts). We also intensively tried to use catalytic amounts of FcBF₄ in combination with a photoredox catalyst. The best result was obtained with {Ir[(dF-CF₃)ppy]₂(dtbpy)}PF₆ (5mol%) and FcBF₄ (20 mol%) (Table 2, entry 8). However, since this method is much less efficient and less attractive than the originally reported method, it was not pursued further. For further details, see also comments to referee 1.

- 2.) In Table 1, when stoichiometric amount of oxidant was applied, reasonable yield was obtained, while with substoichiometric amounts of oxidant, the yield is trace amount (entry 6 vs entry 8). Why?

Answer: We thank Reviewer 3 for this question. In the reactions with substoichiometric amounts of ferrocenium oxidant, one would expect the arylation product to be formed in approximately the same amount as oxidant was added. As Reviewer 3 correctly noted, this is not the case for the reported reaction. We attribute this to the fact that a certain concentration of oxidant must necessarily be present because the enolate is decomposed in a competitive reaction involving intramolecular proton transfer (see also comment to referee 1). This decomposition product (acyl migration product) is observed as the main product in these reactions. This is also the main reason why the development of a catalytic version of the reaction has been so difficult.

- 3.) As is claimed by the authors, a bulky substituent on N atom favors the reaction, why not try t-Butyl sulfonamide?

Answer: We thank the reviewer for this suggestion. We synthesized the requested *tert*-butyl benzenesulfonamide **1g** and used it in our reaction. It turned out that due to its large steric demand, this substrate does not quantitatively react with the ketene to give the corresponding enolate. This also explains the relatively low yield of 33% of the arylated product **3ag**. We have added this example to the revised scheme 2.

REVIEWERS' COMMENTS

Reviewer #1 (Remarks to the Author):

During the previous review, all the three referees pointed out the use of stoichiometric amounts of ferrocenium salt. In this revision, the authors endeavored to improve the reaction by using a catalytic amount of Fe(III), and have obtained a moderate yield with the catalytic conditions. Given the fact that FcBF₄ is inexpensive and commercially available, this method still provides a novel and practical polar-radical-crossover approach. My previous suggestions, such as the scope of heteroaryls and the references, have been carefully responded. Therefore, I supports its publication in NatComm as is.

Reviewer #4:

< In private remarks to the Editor, the reviewer says that the authors' response to concerns of the referees in the previous round was appropriate. >